# Conservation Effectiveness Assessment of the Three Northern Protection Forest Project Area

Yakui Shao [1,2,†] , Yufeng Liu [1,*], Tiantian Ma [1,2,†], Linhao Sun [3,4], Xuanhan Yang [2], Xusheng Li [5] , Aiai Wang [6] and Zhichao Wang [2,*]

1   College of Computer and Information Engineeing, Chuzhou University, Chuzhou 239000, China;
    syk227816_gis@bjfu.edu.cn (Y.S.); matiantian2019@bjfu.edu.cn (T.M.)
2   Precision Forestry Key Laboratory of Beijing, Beijing Forestry University, Beijing 100083, China;
    xuanhanyang@bjfu.edu.cn
3   College of Mathematics and Computer Science, Zhejiang A & F University, Hangzhou 311300, China;
    acesunlh@126.com
4   China Key Laboratory of Forestry Intelligent Monitoring and Information Technology Research of Zhejiang
    Province, Zhejiang A & F University, Hangzhou 311300, China
5   Tianjin Centre of Geological Survey, China Geological Survey, Tianjin 300170, China; saintlxs@foxmail.com
6   School of Geographical Sciences, Harbin Normal University, Harbin 150028, China;
    wangaiai@stu.hrbnu.edu.cn
*   Correspondence: liuyufeng@chzu.edu.cn (Y.L.); wang@forestry.fund (Z.W.)
†   These authors contributed equally to this work.

**Abstract:** The Three-North Shelterbelt Project is the largest ecological engineering initiative in China to date, distinguished by its immense scale, extended construction period, and widespread benefits for the population. The gross ecosystem product (GEP) serves as a crucial indicator for assessing ecological benefits. This study focuses on the Three Northern Protection Forest Project Area, utilizing GEP calculations for the years 2000 to 2020. This study evaluates variations in the production values of different ecosystem services to reflect the ecological conservation benefits of the restoration project. Additionally, it analyzes the spatiotemporal evolution and trends of the GEP calculations, offering data references and decision support for the enduring efficacy of ecological restoration projects. The findings are as follows. (i) Between 2000 and 2020, the GEP of the Three-North region exhibited significant growth with continuous enhancement of various ecosystem service functions; the most substantial rate of change was observed in the water conservation function, followed by carbon sequestration and oxygen release, soil retention, windbreak and sand fixation, flood regulation, and environmental purification functions. (ii) The per-unit area value of different ecosystem types generally increased; the forest ecosystem displayed the largest growth rate at 61.18%, followed by shrubland ecosystems at 49.84%. (iii) The spatial distribution of ecosystem service in the Three-North region displayed a clustering trend alongside notable spatial heterogeneity. High-high clustering zones were identified in areas such as the Tianshan Mountains, Altai Mountains, Qilian Mountains, and Greater and Lesser Khingan Mountains. Conversely, low-low clustering areas were scattered, forming patchy distributions in regions like the Tarim Basin, northern Qinghai-Tibet Plateau, and the Hexi Corridor. This study, by analyzing the gross ecosystem product of the Three-North Shelterbelt Project region, unveils the spatial distribution characteristics, trends, and variations in ecosystem service values over the past two decades. It provides data support and decision guidance for the long-term efficacy of future ecological conservation and restoration projects. This study incorporates the GEP accounting method into the assessment of the effectiveness of major conservation projects. Compared to the traditional methods of effectiveness assessment, this represents a significant exploration and innovation.

**Keywords:** conservation effectiveness; gross ecosystem product; the Three Northern Protection Forest Project Area; spatial autocorrelation

## 1. Introduction

The "Three-North" areas in China encompass the Northwest, North China, and Northeast regions, which are characterized by some of the most challenging natural conditions, including arid or semi-arid climates, water scarcity, and soil erosion. In these regions, the ecological environment has suffered significant degradation, and soil erosion is particularly severe. They are focal points for combating desertification in China [1]. Since the late 20th century, the Chinese government has implemented an integrated afforestation system known as the "Three-North Project" in regions severely affected by sandstorms and significant soil erosion. This project combines the establishment of shelterbelts, forested patches, and vegetation networks with the aim of safeguarding the ecological environment in the Three-North areas [2]. The Three Northern Protection Forest Project Area is one of the largest and longest-running ecological endeavors globally, benefiting a substantial number of people. It has enhanced the ecological environment, effectively addressing wind erosion and sand fixation, soil conservation, and water retention [2,3]. The project has also ensured the well-being of the population by safeguarding their livelihoods [2,3].

Currently, in the field of domestic forestry conservation project assessment, the focus is primarily on the quantitative analysis and evaluation of both ecological and economic benefits. With regard to economic benefits, the main emphasis lies on assessing investment returns and calculating increments in economic output [4,5]. In terms of ecological benefits, the emphasis is placed on calculations related to the benefits of forest ecosystems, among other aspects [6]. In terms of the benefit assessment scale, the primary focus is on small-scale research, often centered around county-level forestry bureaus or forest farms and similar entities [7,8]. Gross ecosystem product (GEP) is calculated to assess and analyze the contributions and roles of ecosystems in human economic and social development as well as their contributions to human well-being. Simultaneously, the accounting of GEP can aid in comprehending and recognizing the status and dynamics of ecosystems [9].

In relevant domestic studies, it has been identified that the gross ecosystem product is a method to quantify the contribution of ecosystems to human economic and social development from the perspective of natural capital [10]. When studying the relationship between ecosystem services and human well-being, both domestic and international literature commonly employ methods such as questionnaire surveys and interviews to conduct qualitative assessments [11]. However, this approach has a certain degree of subjectivity. Therefore, a combined qualitative and quantitative method is necessary for studying the gross ecosystem product (GEP). In the assessment of ecosystem service values, the value of tangible products provided by ecosystems is often referred to as direct use value, while the values of regulating services and cultural services provided by ecosystems are termed indirect use value. In the implementation of the "Two Mountains" theory, the GEP calculation plays a crucial role. The "Two Mountains" theory emphasizes the importance of adhering to the development concept that emphasizes that green mountains and clear waters are as valuable as mountains of gold and silver, suggesting that safeguarding the ecological environment is the most effective path to economic development [11]. Through the GEP calculation, it becomes possible to scientifically quantify the various services that ecosystems provide to the economy and society, such as water conservation, air purification, biodiversity protection, etc. By incorporating the unrealized value of these ecological services into the economic system, a more comprehensive assessment of the sustainability of economic development and the health of the ecological environment can be achieved.

With the implementation of the multi-phase Three-North Shelterbelt Project, it is crucial to comprehend the benefits achieved by the project and its sustainability. A comprehensive and scientifically grounded ecological functional assessment of the Three Northern Protection Forest Project Area is necessary to accurately understand and leverage its significant advantages, thereby guiding effective management. The objectives of this study are as follows: (i) Establish a cost-effective and efficient comprehensive evaluation system for the ecological restoration benefits of the Three-North Shelterbelt Project, aiming to strike a balance between local economic development and environmental/ecological construction;

(ii) Evaluate and analyze the evolving characteristics and trends of the gross ecosystem product (GEP) calculation results within the Three Northern Protection Forest Project Area. This scientific understanding of the development patterns of the Three-North Shelterbelt Project's ecological system will provide a reliable reference for industry planning and policy formulation by government departments at all levels.

## 2. Data and Methodology

### 2.1. General Description of the Study Area and Data Source

As shown in Figure 1, the Three Northern Protection Forest Project Area spans across northeastern China, the northern part of North China, and a significant portion of northwestern China. It encompasses 13 provinces (autonomous regions and municipalities directly under the central government) [12]. The total construction area of the project is 435.8 million square kilometers; the majority of the Three-North region is situated in arid and semi-arid areas that are characterized by a dry climate, decreasing precipitation from east to west, frequent sand and dust storms, and limited water resources [13]. The terrain slopes from west to east, and there is a diverse range of soil types. The vegetation in this area is mainly characterized by semi-arid, arid, and extremely arid plant types, rendering the ecological environment highly fragile [13].

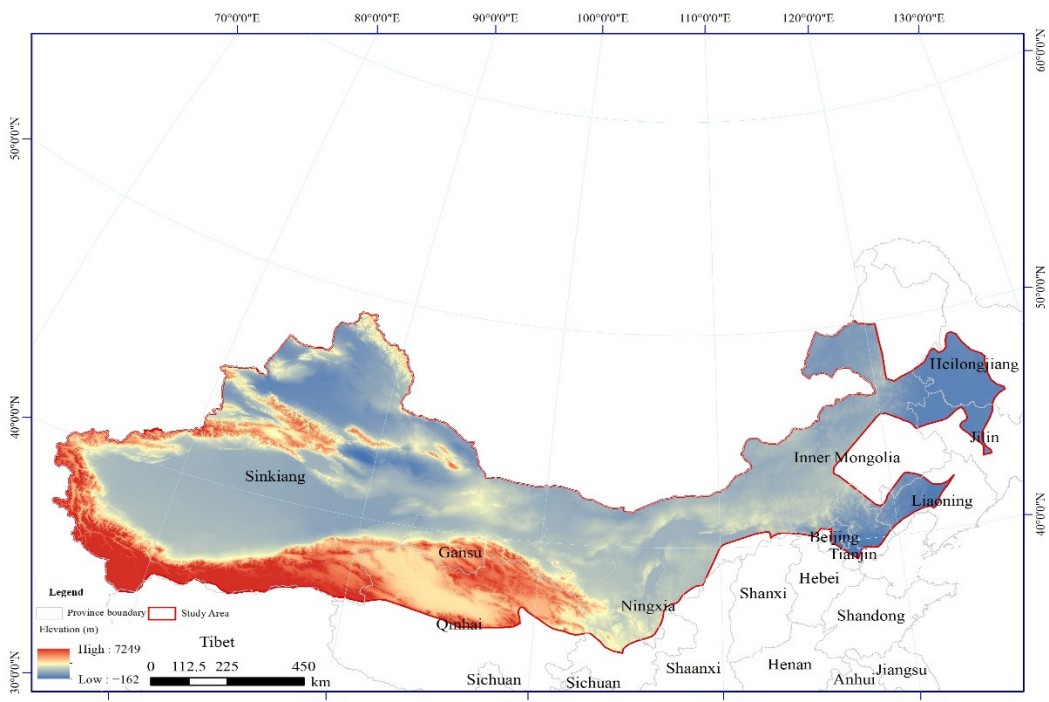

**Figure 1.** Study area.

The data used in this study, as presented in Table 1, were preprocessed to ensure consistency in units and spatial references. Prior to computation, all data underwent standardization. Each data source plays a vital role in the GEP calculations. They provide diverse information that helps us comprehensively understand the value of ecosystem services and connect it to economic activities, thus providing a scientific basis for sustainable development. This involves transforming the coordinate system of raster data into the GCS_WGS_1984 geographic coordinate system. Additionally, the spatial resolution of raster data is resampled to 1 km × 1 km.

**Table 1.** The main data used in this study.

| Classification | Data | Source |
| --- | --- | --- |
| Land use data | Forest/grassland/farmland, etc. | https://www.resdc.cn (accessed on 3 July) |
| Geographic vector | - | https://www.resdc.cn (accessed on 15 July 2020) |
| Vegetation data | Net primary productivity | http://www.ntsg.umt.edu (accessed on 3 May 2022) |
| Statistical data | Statistical yearbook | http://www.stats.gov.cn/ (accessed on 20 September 2021) |
| Meteorological data | Rainfall | http://www.geodata.cn (accessed on 18 May 2021) |

Land use data reflect how different types of land are distributed and utilized in a specific region. Various types of land, such as cropland, forests, and grasslands, offer different ecosystem services and economic values. Consequently, land use data serve as the cornerstone for assessing the value of ecosystem services. NPP is a crucial metric when it comes to evaluating the worth of ecosystem services that are associated with climate regulation, carbon sequestration, and oxygen production. It is intricately tied to ecosystem productivity and carbon cycling. Therefore, NPP is an essential data source for calculating the GEP.

Statistical yearbooks encompass a wealth of data related to the economy, society, and the environment. They offer insights into regional economic activities, GDP, employment, and more, serving as crucial resources for assessing the contribution of ecosystem services to the economy. By leveraging statistical yearbook data, it becomes possible to link the economic value of ecosystem services to local economic activities, providing valuable information for the development of environmental policies. Climate data are a fundamental component for assessing ecosystem service values and directly influence various services, like water resource supply, crop yields, and water resource regulation. Climate data can also be used to model the impact of climate change on ecosystem services.

In summary, each data source plays a pivotal role in the GEP calculations. These sources provide a diverse range of information that allows for us to comprehensively grasp the value of ecosystem services and establish connections with economic activities. This, in turn, offers a scientific foundation for sustainable development.

*2.2. The Evaluation of the Gross Ecosystem Product*

The research roadmap for this study is depicted in Figure 2. We developed a comprehensive index system for the assessment of the ecosystem gross production value in the Three-North region. This system utilizes a multitude of data sources, including land usage, net primary productivity (NPP) of vegetation, climate and meteorological data, and statistical yearbooks. We employed this system to calculate the ecosystem gross production value from 2000 to 2020. Furthermore, we analyzed the spatiotemporal evolution patterns and trends in the calculated results of the ecosystem gross production value.

In Table 2, drawing upon previous research [14–17] and on-site investigations, we developed a comprehensive index system for the gross ecosystem product (GEP) calculation within the Three-North Shelterbelt Project area.

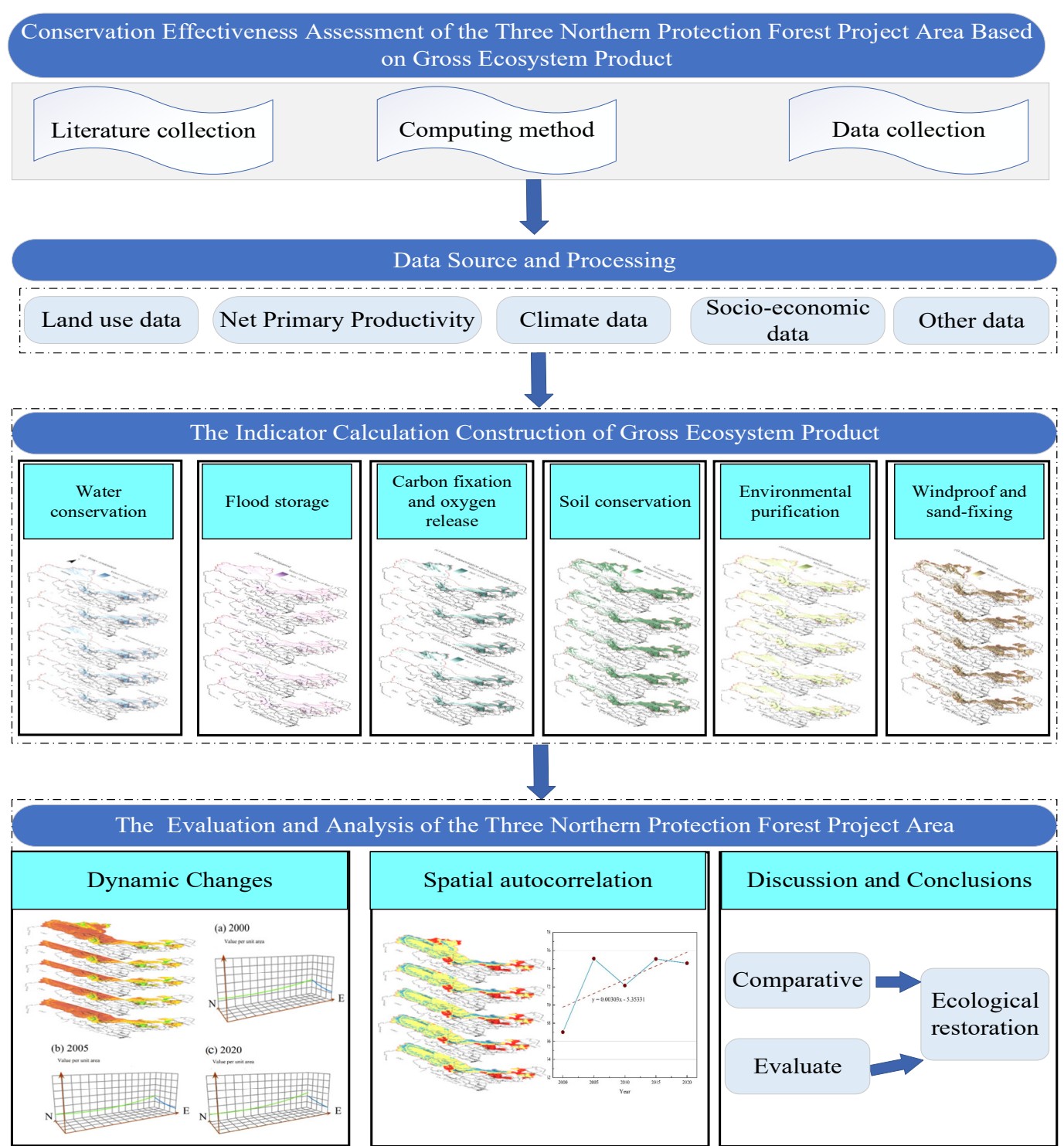

**Figure 2.** Diagram illustrating the approach employed in this research.

**Table 2.** Ecosystem service evaluation index used in this study.

| Calculation Method | Formula | Reference |
|---|---|---|
| Water conservation value | $$V_w = C_w \cdot \sum_{i=1}^{l} \sum_{j=1}^{m} \sum_{k=1}^{n} \left( A_{ijk} \cdot J_i \cdot R_j \cdot K_k \right)$$ $V_w$ represents the value of water conservation services, $A_{ijk}$ represents the area of the study area, $J_i$ is the annual average precipitation, $R_j$ is the benefit coefficient of reduced runoff compared to bare land, and $C_w$ is the engineering cost per unit storage capacity of a reservoir. | [14–17] |
| Flood storage value | $$V_h = (G_{h1} + G_{h2}) \cdot C_w$$ $V_h$ represents the value of water conservation services, $G_{h1}$ is the adjustable flood storage capacity of lakes, $G_{h2}$ is the flood control storage capacity of reservoirs, and $C_w$ is the engineering cost per unit storage capacity of a reservoir. | [17,18] |
| Carbon fixation and oxygen release value | $$V_f = \sum_{i=1}^{j} NPP_j \cdot (1.62 \cdot C_c) + \sum_{i=1}^{j} NPP_j \cdot (1.2 \cdot C_q)$$ $V_f$ represents the total value of carbon sequestration and oxygen release, $NPP_j$ stands for the net primary productivity of ecological asset type $j$, $C_c$ is the price of carbon sequestration in the market, and $C_q$ is the price of oxygen production in the market. | [19] |
| Soil conservation value | $$V_k = V_{k1} + V_{k2} + V_{k3}$$ $V_k$ represents the total value of soil conservation, $V_{k1}$ is the economic benefit of preserving soil fertility, $V_{k2}$ is the economic benefit of reducing land abandonment, and $V_{k3}$ is the economic benefit of reducing sediment accumulation disasters. | [17] |
| Environmental purification value | $$V_n = \sum_{i=1}^{j} G_i \cdot A \cdot C_{ni}$$ $V_n$ represents the total value of water quality purification, $G_i$ is the unit area purification quantity of water pollutant type $i$, $A$ is the wetland area, and $C_{ni}$ is the treatment cost of water pollutant type $i$. $$V_h = \sum_{i=1}^{n} \left( G_h \cdot C_i \right)$$ $V_h$ represents the total value of air purification, $G_h$ is the total mass of air pollutants purified, and $C_i$ is the treatment cost of various types of air pollutants. | [20] |
| Windproof and sand-fixing value | $$V_s = G_S / (\rho \cdot h) \cdot C_s$$ $V_s$ represents the value of windbreak and sand fixation, $G_S$ is the total mass of windbreak and sand fixation materials, $\rho$ is the soil bulk density, $h$ is the thickness of soil surface sand coverage, and $C_s$ is the average cost of sand control engineering. | [17,21,22] |

The importance of these ecosystem service indicators lies in their direct connection to the survival, development, and well-being of human society, forming the foundation of the interdependence and mutual promotion between ecosystems and the socio-economic system.

### 2.3. Spatial Autocorrelation of the Three-North Shelterbelt Project Area

Spatial autocorrelation analysis is a statistical method employed for studying geographical spatial data. It comprises two categories: global spatial autocorrelation (Global Moran's I) and local spatial autocorrelation (Local Indicators of Spatial Association, LISA) [23]. The formula is as follows [24]:

Global autocorrelation model:

$$I = \frac{\sum_{i=1}^{n} \sum_{j=1}^{n} w_{ij}(x_i - \bar{x})(x_j - \bar{x})}{S^2 \sum_{i=1}^{n} \sum_{j=1}^{n} w_{ij}}$$

Local autocorrelation model:

$$I_i = \frac{(x_i - \bar{x})}{S^2} \sum_{j=1}^{n} (x_j - \bar{x})$$

In the equation, $n$ represents the sample size; $S^2$ denotes the sample variance; $x_i$ and $x_j$, respectively, represent the observed values of the variable $x$ in spatial geographic units $i$ and $j$; $\bar{x}$ signifies the mean value of the variable x under study; and $w_{ij}$ stands for the spatial weight matrix, revealing the spatial relationships between each unit [25]. Spatial analysis involves identifying significant hot spots and cold spots, which are phenomena or events in geographic space that exhibit significantly higher or lower values than the average. This analysis is commonly used to identify concentrated or dispersed patterns within a specific geographical area. To calculate confidence values for hotspots and coldspots, spatial statistical methods are typically employed. One commonly used method is the Getis–Ord Gi* statistic, also known as hotspot analysis or G-statistics. Statistical significance is determined using a $p$-value (significance level) with conclusions drawn if the $p$-value is less than the chosen significance level, typically 0.05.

Moran's index and local Moran's I assist in assessing the presence of significant spatial patterns in the distribution of the GEP, shedding light on trends and helping policymakers make informed decisions. For instance, if high-value clusters are detected in specific regions, governments may consider implementing environmental conservation or resource management policies there. Conversely, negative Local Moran I values may signal the need for targeted interventions in particular areas to address potential concerns.

This study employed the Anselin Local Moran I tool in ArcGIS 10.7 and overlaid land use change maps.

## 3. Results

### 3.1. Current Status of the GEP in the Study Area in 2020

From the perspective of the gross ecosystem product (GEP), as shown in Table 3, among the six ecosystem services, carbon sequestration and oxygen release held the highest functional value, accounting for 43.71% and representing a core service function of the Three-North ecosystem. Soil retention followed as the second highest, constituting 26.24%. Subsequently, windbreak and sand fixation, water conservation, flood regulation, and environmental purification account for 17.03%, 9.93%, 2.18%, and 0.9%, respectively. Regarding ecosystem types, grassland ecosystems held the highest value at 1378.266 billion yuan, representing 68.81% of the total, while farmland ecosystems followed with 338.983 billion yuan, constituting 16.92%. Forest, shrubland, and aquatic ecosystems had comparatively lower values, amounting to 150.788 billion yuan, 76.625 billion yuan, and 58.343 billion yuan, respectively, and representing 7.53%, 3.83%, and 2.91%, respectively. Regarding per-unit area values, shrubland ecosystems exhibited the highest value at 168.5 thousand yuan/km$^2$, closely followed by forest ecosystems at 165.68 thousand yuan/km$^2$. Subsequently, grassland and farmland ecosystems had per-unit area values of 121.03 thousand yuan/km$^2$ and 101.59 thousand yuan/km$^2$, respectively. Lastly, aquatic ecosystems had the lowest per-unit area value at 83.48 thousand yuan/km$^2$.

Examining the spatial distribution of the GEP for the current year (Figure 3), the distribution in 2020 generally displayed high values in the eastern and northwestern parts with a lower pattern in the central area. This transition signifies an increase from the central area that is dominated by desert ecosystems with limited precipitation towards the eastern and northwestern regions with predominant grassland, aquatic, shrubland, and forest ecosystems that receive higher rainfall. The regions with higher values for each service type correlated closely with the spatial distribution of forest, shrubland, grassland, and farmland ecosystems.

**Table 3.** 2020 GEP accounting results in the Three Northern Regions of China.

| Types | Value (Hundred Million Yuan) | | | | | |
|---|---|---|---|---|---|---|
| | Forest | Shrub | Grassland | Farmland | Water | Total |
| Water retention | 634.67 | 208.61 | 782.68 | 363.39 | / | 1989.36 |
| Flood mitigation | / | / | / | / | 436.97 | 436.97 |
| Carbon sequestration and Oxygen production | 510.93 | 243.06 | 6558.88 | 1442.34 | / | 8755.22 |
| Soil retention | 307.29 | 182.17 | 3653.76 | 1113.31 | / | 5256.53 |
| Environment purification | 25.79 | 5.23 | 16.36 | / | 133.18 | 180.56 |
| Sandstorm prevention | 29.19 | 127.17 | 2770.97 | 470.80 | 13.28 | 3411.41 |
| Total | 1507.88 | 766.25 | 13782.66 | 3389.83 | 583.43 | 20,030.05 |

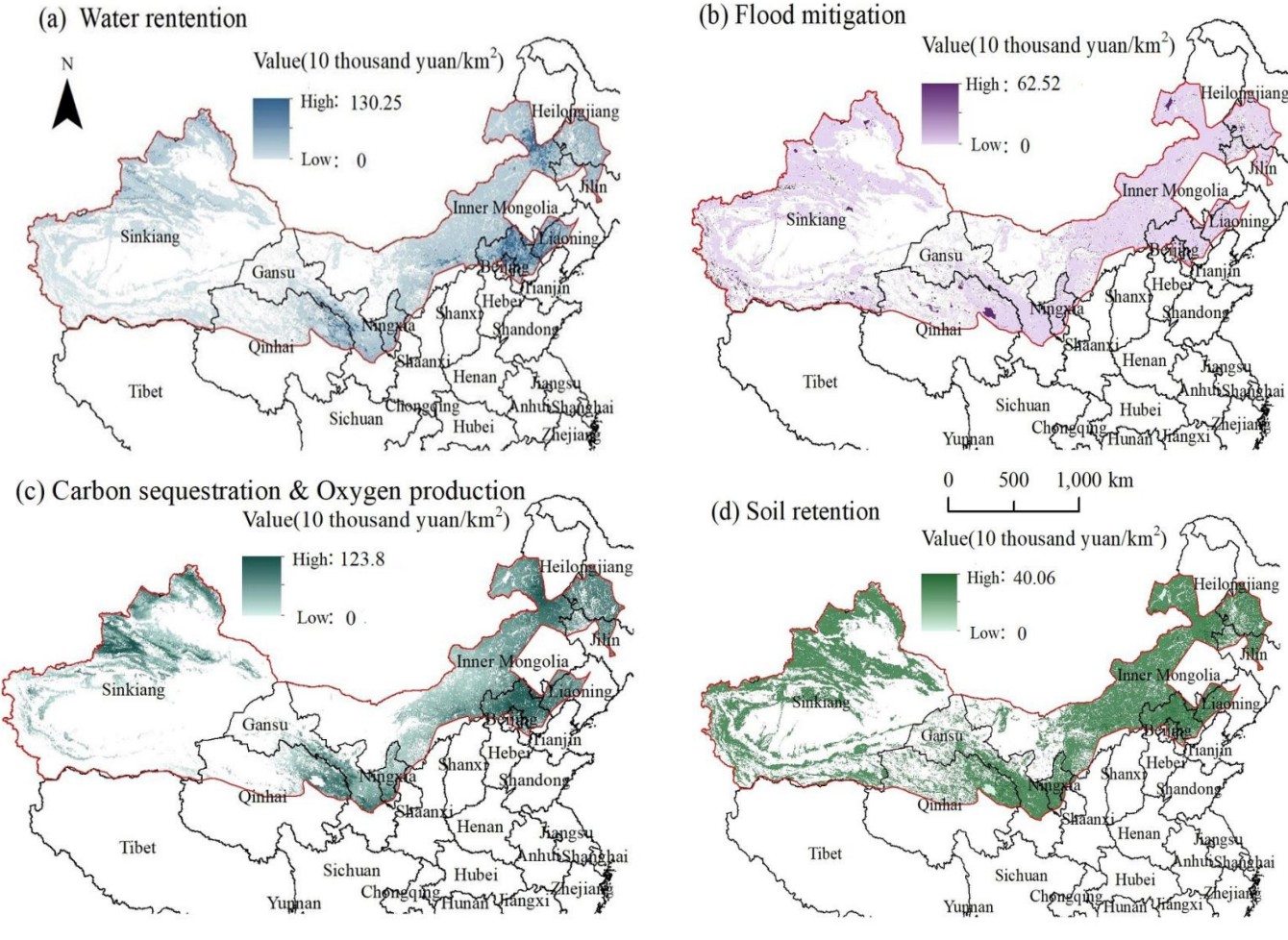

**Figure 3.** *Cont.*

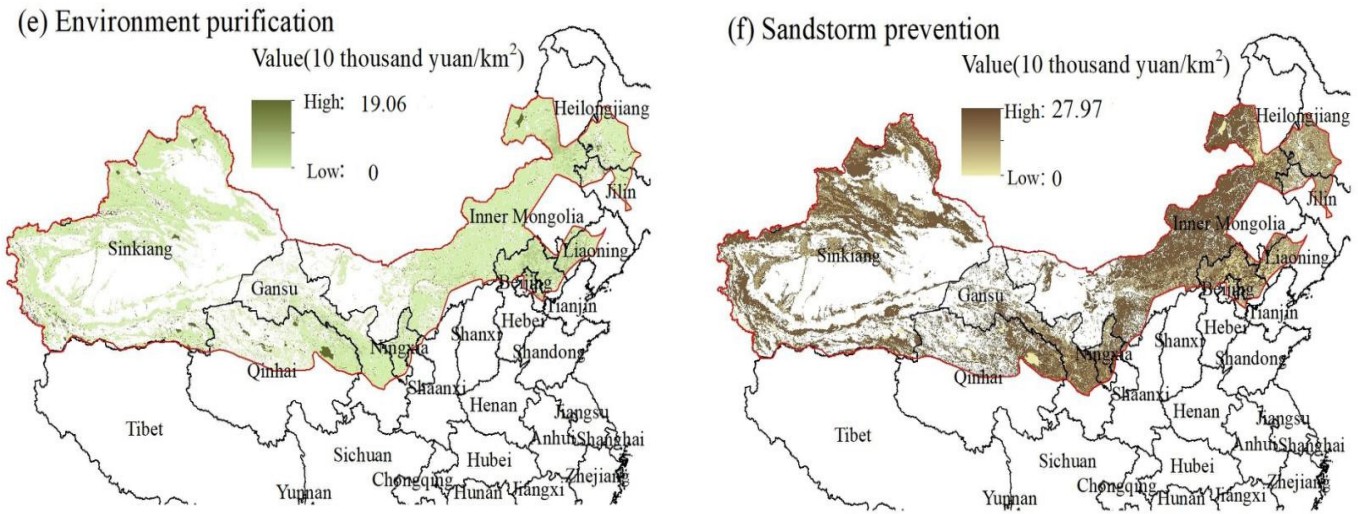

**Figure 3.** Distribution map of the GEP by function of the Three-North region in 2020.

*3.2. Dynamic Changes in the Gross Economic Product in the Three-North Region from 2000 to 2020*

In terms of the gross ecosystem product (GEP) analysis (Table 4), excluding price factors and using comparable prices, the Three Northern Protection Forest Project Area exhibited a clear overall growth trend from 2000 to 2020. The GEP increased from 1446.976 billion yuan in 2000 to 2003.005 billion yuan in 2020, indicating a growth of 38.43%. Looking at the values of various service functions (Table 2), over the course of two decades, all service functions demonstrated growth trends. Among them, water conservation showed the highest rate of change, reaching 69.17%. Tree planting and afforestation, wetland preservation, and policies prohibiting overlogging and land degradation have contributed to enhancing water conservation functions and promoting the protection and augmentation of water resources. These measures help maintain soil water retention, thus elevating water conservation functions. Carbon sequestration and oxygen release ranked second with a change rate of 42.67%. Soil retention achieved a change rate of 31.58%, placing third. Subsequently, windbreak and sand fixation, flood regulation, and environmental purification demonstrated change rates of 30.73%, 12.41%, and 10.17%, respectively.

**Table 4.** Accounting results of the GEP in the Three Northern Regions of China from 2000 to 2020.

| Types | Value (Million) | | | | | Change Rate (%) |
|---|---|---|---|---|---|---|
| | **2000** | **2005** | **2010** | **2015** | **2020** | **2000–2020** |
| Water retention | 1175.94 | 1525.46 | 1912.40 | 1710.95 | 1989.36 | 69.17 |
| Flood mitigation | 388.74 | 425.50 | 497.25 | 477.47 | 436.97 | 12.41 |
| Carbon sequestration and Oxygen production | 6136.71 | 6898.38 | 8118.21 | 7893.95 | 8755.22 | 42.67 |
| Soil retention | 3995.05 | 4392.82 | 5140.96 | 4894.56 | 5256.53 | 31.58 |
| Environment purification | 163.89 | 179.35 | 209.52 | 201.39 | 180.56 | 10.17 |
| Sandstorm prevention | 2609.42 | 2861.14 | 3346.44 | 3178.27 | 3411.41 | 30.73 |
| Total | 14,469.76 | 16,282.65 | 19,224.78 | 18,356.59 | 20,030.05 | 38.43 |

Analyzing the per-unit area values across the different ecosystems (Table 5), the overall per-unit area value increased from 86.5 thousand yuan/km$^2$ in 2000 to 119.28 thousand yuan/km$^2$ in 2020, representing a growth of 37.9%. Across ecosystem types, the per-unit area values generally exhibited an upward trend. Among them, forest ecosystems witnessed the highest increase, reaching 61.18%. Shrubland ecosystems followed with an increase of 49.84%. Subsequently, the per-unit area values for grassland, aquatic, and farmland ecosystems increased by 37.67%, 36.36%, and 27.82%, respectively. Notably, grassland ecosystems exhibited a relatively smaller change in the per-unit area value but maintained

a contribution of over 68.81% to the GEP for many years, indicating its substantial local impact.

**Table 5.** Accounting results of the GEP per unit area in the Three Northern Regions of China from 2000 to 2020.

| Types | Value per Unit Area (ten thousand/km$^2$) | | | | | Change Rate (%) |
|---|---|---|---|---|---|---|
| | 2000 | 2005 | 2010 | 2015 | 2020 | 2000–2020 |
| Forest | 102.79 | 131.12 | 155.11 | 145.11 | 165.68 | 61.18 |
| Shrub | 112.46 | 136.10 | 161.42 | 153.15 | 168.50 | 49.84 |
| Grassland | 87.91 | 99.63 | 117.62 | 112.49 | 121.03 | 37.67 |
| Farmland | 79.40 | 81.29 | 95.56 | 91.26 | 101.49 | 27.82 |
| Water | 61.22 | 67.38 | 78.77 | 74.62 | 83.48 | 36.36 |
| Total | 86.50 | 97.34 | 114.80 | 109.39 | 119.28 | 37.90 |

In the trend surface analysis of spatial distribution (Figures 4 and 5), the results showed that, from 2000 to 2010, the fitted curves gradually declined in both east–west and north–south directions, indicating a decreasing per-unit area value. However, from 2010 to 2020, the fitted curves in both directions gradually increased, indicating a rising per-unit area value. Furthermore, the per-unit area value for 2020 was significantly higher than that of 2000, presenting an overall increasing trend over the 20-year period.

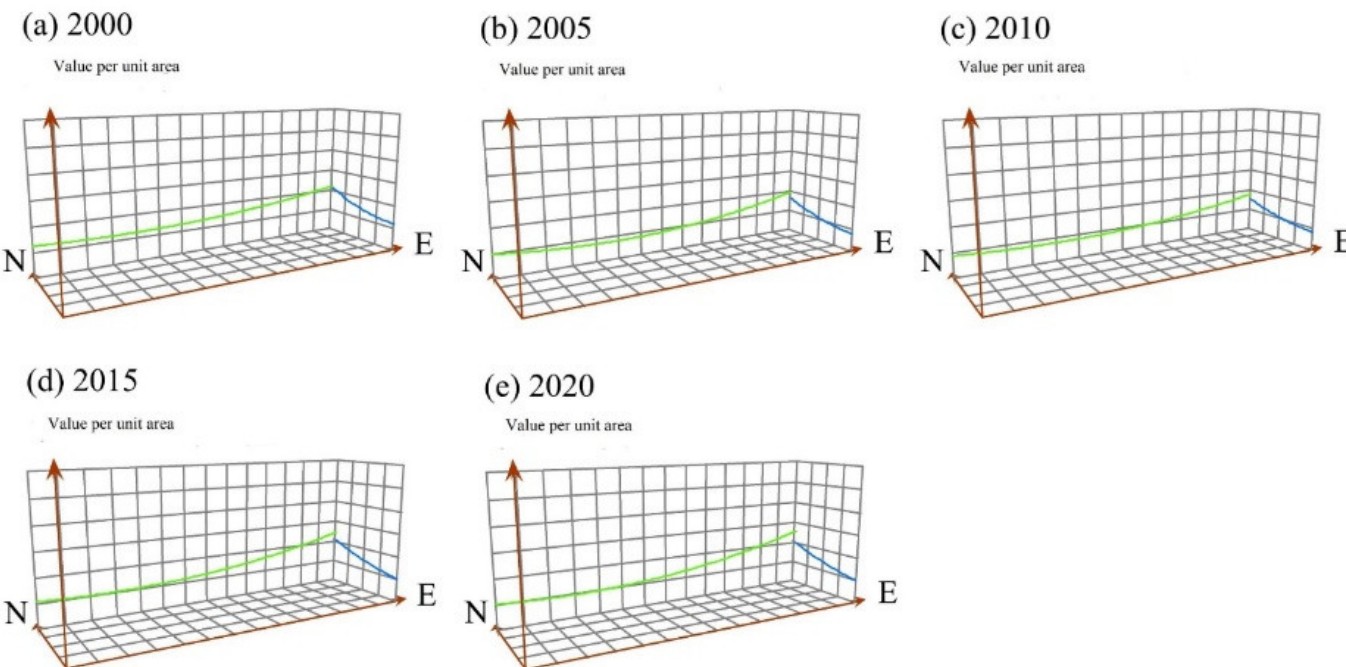

**Figure 4.** Trend of the GEP per unit area in the Three Northern Regions of China from 2000 to 2020 (the N-axis represents the true north direction, and the E-axis represents the true east direction. The green line represents the east–west trend, while the blue line represents the north–south trend. The Z value represents the value of ecosystem service function per unit area in the corresponding year.

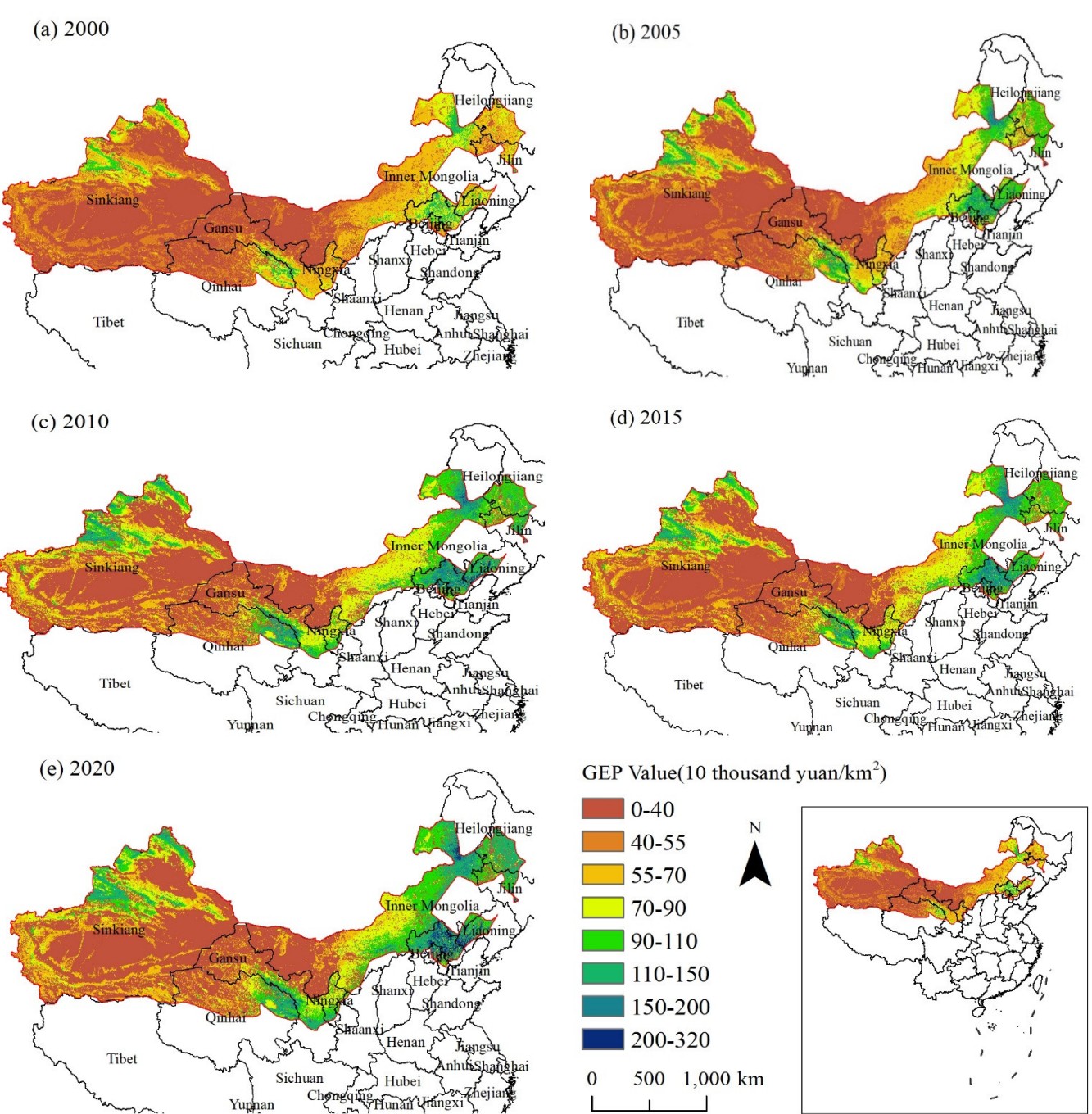

**Figure 5.** Spatial distribution of the GEP per unit area in the Three Northern Regions of China from 2000 to 2020.

### 3.3. Spatial Autocorrelation Analysis of the Three-North Region

From the Moran's index scatter plot (Figure 6), it can be observed that the global Moran's index for the years 2000, 2005, 2010, 2015, and 2020 was 0.725, 0.777, 0.768, 0.780, and 0.780, respectively. The overall trend indicates a gradual increase, signifying an elevation in the spatial clustering of ecosystem service values within the Three-North region. All the indices are greater than 0, indicating that there is positive spatial autocorrelation in the gross ecosystem product across the study area. The rising trend in the Moran's index signifies an intensifying spatial clustering of the GEP values within the Three-North region. This underscores the necessity for customized policies and calls for in-depth exploration,

including a thorough consideration of the spatial aspect in resource management and conservation initiatives.

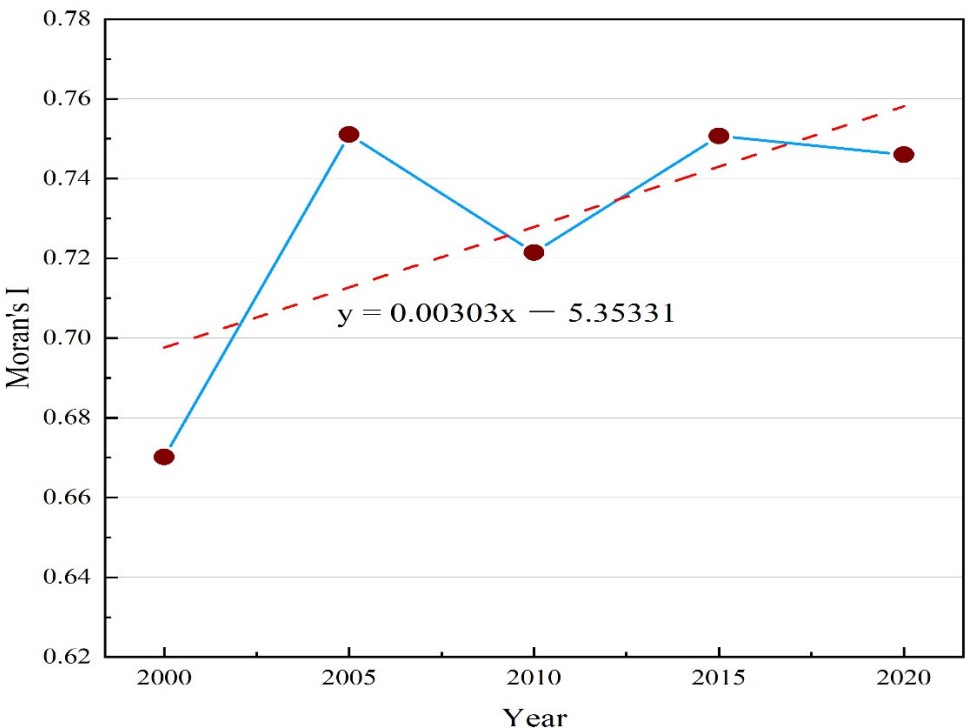

**Figure 6.** Change in the global Moran's index from 2000 to 2020.

A local indicators of spatial association (LISA) analysis of ecosystem service values in the Three-North region was conducted (Figure 7). Spatially, the local Moran's index demonstrated a strong low-high clustering in each phase, revealing notable spatial outliers. Throughout the study period, the spatial distribution of the high-high and low-low clusters remained consistent. The high-high clusters were characterized by extensive aggregation, predominantly located in regions with concentrated forest and shrubland ecosystems, such as the Tianshan Mountains, Altai Mountains, Qilian Mountains, Greater and Lesser Khingan Mountains, as well as the northern part of Hebei Province to the western part of Liaoning Province. This alignment corresponds to the higher per-unit area values associated with forest and shrubland ecosystems. In contrast, the spatial distribution of the low-low clusters was dispersed and patchy, found in areas like the Tarim Basin, Junggar Basin, transition zones around the Turpan Depression and surrounding mountain ranges, northern Tibetan Plateau, Hexi Corridor, and the interior of the Inner Mongolian Plateau along the Wei River. This distribution corresponds to the lower per-unit area values of grassland and farmland ecosystems.

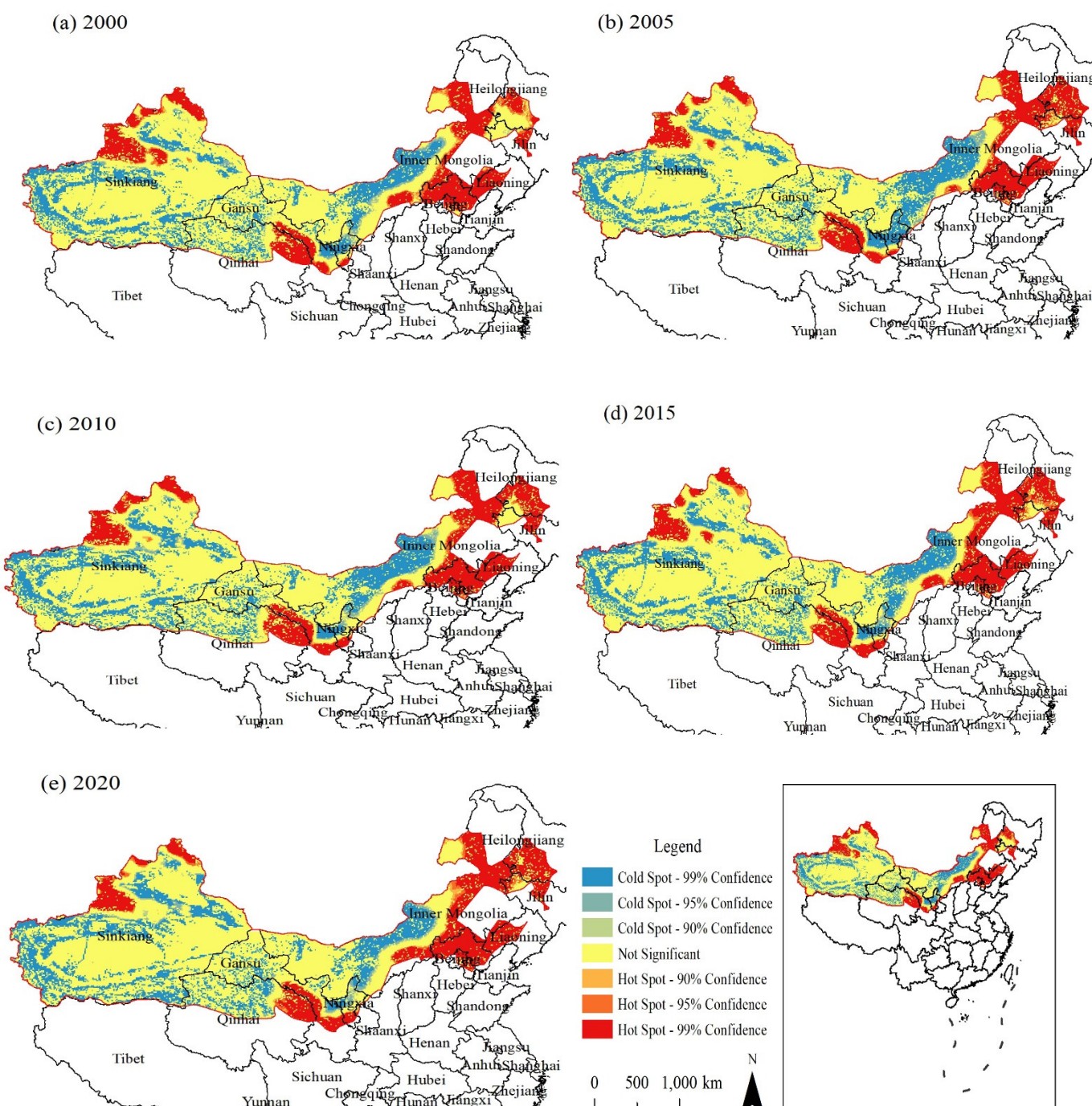

**Figure 7.** Distribution of the GEP hot spots (cold spots) in the Three Northern Regions of China from 2000 to 2020.

## 4. Discussion and Conclusions

### 4.1. Discussion

The value of ecosystem services can vary due to geographic location, scale, and spatial scope. Depending on the assessment's objectives, it is important to select an appropriate spatial scale and accounting precision [9]. The issues in the construction of the "Three-North" Shelterbelt Project are an insufficient actual investment of construction funds and degradation of afforestation projects [26]. The growth, stability, or decline of the gross ecosystem product reflects the evolutionary trend of ecosystems in supporting economic and social development [9]. Valuation involves translating the benefits of ecosystem

services into comprehensible concepts, but it does not necessarily require converting them into monetary units [27]. The initial intention of the GEP (economic accounting of natural capital) is to enhance the conservation of natural capital. GEP accounting should not be directly regarded as the actual representation of ecosystem service values in the market; it primarily reflects the ecological service values of natural capital [28]. Therefore, by accounting for the gross ecosystem product, the level and condition of the Three Northern Protection Forest Project Area's sustainable development can be assessed, and the effectiveness of its ecological conservation measures can be evaluated.

This study has several limitations. First, due to the diverse array of data types, we encountered challenges in obtaining certain data, and the parameters available for calculation were relatively limited. Second, there is a great diversity of ecosystem service types with many yet to be fully explored. Even among the known service types, there are issues with imperfect calculation methods and a lack of standardized parameters. Therefore, the accounting indicators in this study have not been able to fully encompass all ecosystem service functions within the study area. Previous studies [29] have indicated that the Three-North Shelterbelt Project has, on the whole, yielded significant ecological benefits from 2001 to 2020 (sharing common trends with this study). However, the difference lies in the fact that this study employs the GEP (gross ecosystem product) ecosystem production value accounting index system, encompassing the ecosystem macro-structure change index (EMSCI), quality change index (EQCI), service function change index (ESCI), and ecosystem recovery index (ERI). The strength of this study lies in the utilization of the GEP ecosystem production value accounting index system, providing a robust tool for a more comprehensive assessment of the ecological benefits of the Three-North Shelterbelt Project. The construction of the GEP indicator system for this study was inspired by previous research and work [14–17]. This index system offers a more comprehensive reflection of ecosystem quality, structure, service functions, and recovery status, enabling us to gain deeper insights into the changing trends and ecological benefits of the ecosystem. This approach enhances the scientific rigor and practical applicability of the research, offering a more comprehensive dataset and insights for future ecological conservation and sustainable development.

When quantifying ecosystem services in regions with pronounced spatial heterogeneity, there is uncertainty in the utilization of the average values of ecosystem services for various land-use types [30]. Different types of ecosystems exhibit variations in their functions and structures; for instance, forests, wetlands, and grasslands provide ecosystem services that differ due to their specific characteristics. Human activities also play a significant role in shaping land use and impacting ecosystems. Activities like agriculture, urbanization, and industrialization can alter land use types and ecosystem functions, consequently affecting the provisioning of ecosystem services. It is necessary to establish unified GEP accounting standards and regulations in our country to avoid contradictions and limitations between different accounting methods [31].

As a comprehensive ecological value indicator, the variations in the GEP can be directly employed to substantiate the assessment of the effectiveness of conservation measures, thereby offering a novel approach for assessing the effectiveness of similar conservation projects. In future research, we should adopt diverse research methodologies and focus on the following aspects. First, we emphasize the diversity of the data sources because the evaluation of ecosystem services typically requires the inclusion of multiple data types and origins. In broad ecosystem assessments, we may need to use general data to gain a wide-ranging overview. However, for more detailed local assessments, finer and more precise data are required to better understand the characteristics of the local ecosystems. Therefore, the diversity of the data sources is aimed at ensuring that our research is not only broadly applicable but also rich in detailed information [27]. Second, multi-scale research is chosen because the characteristics of ecosystem services and ecosystems themselves exhibit significant variations across different spatial and temporal scales. We opt for a multi-level analysis, spanning from micro to macro perspectives, to comprehensively understand the

ecological evolution and developmental trends within the Three Northern Protection Forest Project Area. Specifically, we focus on how the evolution of societal human factors has led to an increase in land use intensity, resulting in the gradual transformation of natural ecosystems into semi-natural, semi-artificial, and even fully artificial ecosystems. This transformation affects the flow of matter and energy, consequently influencing the capacity of ecosystems to provide services. Therefore, multi-scale research is instrumental in gaining a better understanding of these complex interactions [32]. Furthermore, it is crucial to emphasize the diversity of ecosystem services and consider the varying contributions of different ecosystems to supporting economic and societal development. This approach can provide scientific insights to formulate more comprehensive and targeted regional development and ecological conservation strategies.

In summary, we have chosen these future research directions to better address the complexity of our study and the challenges that lie ahead. These directions are selected with good reason and are aimed at providing in-depth research and practical guidance for the development and ecological conservation of the Three Northern Protection Forest Project Area. The choice of these directions is grounded in the aim of enhancing the scientific quality and practical applicability of our research.

*4.2. Conclusions*

This research centers on the Three Northern Protection Forest Project Area, employing GEP computations that span the period from 2000 to 2020. This study assesses the variances in the production values of diverse ecosystem services to portray the ecological preservation benefits of the restoration initiative. Furthermore, it scrutinizes the spatiotemporal evolution and tendencies in the GEP calculations, offering data references and decision-making support for the enduring effectiveness of ecological restoration undertakings. The findings reveal the following.

(i) During the interval from 2000 to 2020, the GEP of the Three Northern region exhibited notable expansion, accompanied by sustained enhancements in various ecosystem service functions. The most remarkable rate of change was noted in the water conservation function, followed by carbon sequestration, oxygen release, soil retention, windbreak, sand fixation, flood regulation, and environmental purification functions.

(ii) The per-unit area value of distinct ecosystem categories generally experienced an increase. Notably, the forest ecosystem demonstrated the highest growth rate at 61.18%, closely trailed by shrubland ecosystems at 49.84%.

(iii) The spatial distribution of ecosystem services across the Three Northern region displayed a clustering pattern along with conspicuous spatial heterogeneity. Regions exhibiting high-high clustering zones were identified in areas such as the Tianshan Mountains, Altai Mountains, Qilian Mountains, and Greater and Lesser Khingan Mountains. Conversely, regions displaying low-low clustering characteristics were scattered, resulting in fragmented distributions across regions like the Tarim Basin, northern Qinghai-Tibet Plateau, and the Hexi Corridor. The analysis of the gross ecosystem product within the Three Northern Shelterbelt Project region unveils the spatial distribution attributes, trends, and fluctuations in ecosystem service values over the past two decades. It furnishes data reinforcement and decision-oriented guidance for the enduring effectiveness of future ecological conservation and restoration initiatives. This research seamlessly integrates the GEP accounting approach into the evaluation of major conservation endeavors. In comparison to the conventional methods of effectiveness assessment, this marks a substantial exploration and innovation.

**Author Contributions:** Conceptualization, Y.L., Z.W., Y.S. and T.M.; data curation, Y.S., Z.W., T.M. and Y.L.; formal analysis, Y.S., T.M., Z.W. and Y.L.; funding acquisition, Y.L.; investigation, Y.S., T.M., Z.W. and Y.L.; methodology, Y.L., Y.S., T.M. and Z.W.; project administration, Y.L. and Y.S.; resources, Y.L. and Y.S.; supervision, Y.S. and Y.L.; validation, Y.L., Y.S., L.S., X.Y., T.M., Z.W. and A.W.; visualization, Y.S. and Y.L.; writing—original draft, Y.S. and Y.L.; writing—review and editing, Y.S.,

Y.L., L.S., X.Y., T.M., Z.W., X.L. and A.W. All authors have read and agreed to the published version of the manuscript.

**Funding:** This research was funded by the major national R&D project of high-resolution Earth observation system (76-Y50G14-0038-22/23), Major Science and Technology Projects in Anhui Province (202003a06020002), Natural Science Research Project of Universities in Anhui Province (KJ2021A1063), and Science and Technology Plan Project of Chuzhou City (2021ZD013).

**Data Availability Statement:** The data supporting the results of this study can be obtained by contacting the corresponding author.

**Acknowledgments:** We would like to thank the editors and reviewers for their valuable opinions and suggestions that improved this research.

**Conflicts of Interest:** The authors declare no conflict of interest.

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
