# Peer review of "Conservation Effectiveness Assessment of the Three Northern Protection Forest Project Area"

_forests, doi:10.3390/f14112121_

Round 1

Reviewer 1 Report

While the introduction provides a general background, it would be helpful to specify the key environmental issues, such as the extent of soil erosion, desertification, and their ecological and societal implications. This would give readers a more detailed understanding of the challenges faced.

It might be beneficial to include some key statistics about the area, such as the total land area covered by the Three Northern Protection Forest Project and its population.

Resolution of the Figures in the manuscript should be significantly increased to improve the readability.

Equations shouldn’t be inserted as images, but be written using LaTeX or Word MathType

In Table 1, you've listed the main data sources used in the study, which is important for transparency. However, it would be helpful to briefly mention how each data source was collected or obtained. For example, was the meteorological data collected from weather stations or remote sensing sources?

When discussing the index system for GEP calculation, you mention that it uses multiple data sources, including land usage, net primary productivity, climate and meteorological data, and statistical yearbooks. It would be beneficial to briefly explain why each of these data sources is essential for GEP calculation. For example, how does net primary productivity relate to GEP, and why is climate data relevant?

In the sentence "Building upon previous research and on-site investigations," it would be helpful to provide a brief reference to or citation of the relevant previous research.

It might be beneficial to provide a brief rationale for why spatial autocorrelation analysis is relevant to your study. How does it contribute to understanding the spatial patterns of GEP or other aspects of the research?

The analysis of temporal changes in GEP is valuable. However, some explanations of the findings would enhance the reader's understanding. For instance, you mention that "water conservation showed the highest rate of change, reaching 69.17%." Providing some context as to why this occurred or its implications for the environment would be beneficial.

Ensure consistency in units throughout the tables. In Table 4, the unit "Value(million)" is not a standard unit of measurement, and it should be clarified.

While the use of Moran's Index and Local Moran I is appropriate, it's important to explain their practical significance. How do the values of these indices inform your study, and what do they suggest about the spatial distribution of GEP?

In Figure 6, where you present the change in the global Moran's index, it would be beneficial to clarify the significance of the increasing trend. Is this trend expected, and what does it imply for the distribution of GEP in the region?

I wish that my comment would be helpful in improving the quality of this research.

Thank you.

Author Response

While the introduction provides a general background, it would be helpful to specify the key environmental issues, such as the extent of soil erosion, desertification, and their ecological and societal implications. This would give readers a more detailed understanding of the challenges faced.

It might be beneficial to include some key statistics about the area, such as the total land area covered by the Three Northern Protection Forest Project and its population.

Response: Thanks for your proposal. It has been added.  (See Page 3, Line 114-115;)

Resolution of the Figures in the manuscript should be significantly increased to improve the readability.

Response: Thanks for your proposal. It has been changed. (See Page 3, Line120-122;)

Equations shouldn’t be inserted as images, but be written using LaTeX or Word MathType

Response: Thanks for your proposal. It has been changed. (See Page 5, Line141-143;)

In Table 1, you've listed the main data sources used in the study, which is important for transparency. However, it would be helpful to briefly mention how each data source was collected or obtained. For example, was the meteorological data collected from weather stations or remote sensing sources?

Response: The data for the meteorological station is sourced from remote sensing data, with the data source provided on the website. (See Page 3-4, Line127-128;)

When discussing the index system for GEP calculation, you mention that it uses multiple data sources, including land usage, net primary productivity, climate and meteorological data, and statistical yearbooks. It would be beneficial to briefly explain why each of these data sources is essential for GEP calculation. For example, how does net primary productivity relate to GEP, and why is climate data relevant?

Response: Various data sources play a crucial role in GEP (Gross Ecosystem Product) calculations because they provide different dimensions and indicators for assessing the value of ecosystem services. The importance of each data source in GEP calculations is as follows:

(1) Land Use Data:

Importance: Land use data reflects the distribution and utilization of different types of land within a specific region. Different types of land, such as cropland, forests, grasslands, etc., provide different ecosystem services and economic values. Therefore, land use data forms the foundation for assessing the value of ecosystem services.

(2) Net Primary Productivity (NPP):

Relationship: NPP refers to the rate at which plants absorb carbon dioxide from the atmosphere through photosynthesis and convert it into organic matter. It is a fundamental productivity indicator of ecosystems and a significant component of ecosystem service value, directly reflecting the ecosystem's contribution to carbon sequestration and oxygen production.

Importance: NPP is a crucial indicator in assessing ecosystem service values related to climate regulation, carbon sequestration, and oxygen production. It is directly linked to ecosystem productivity and carbon cycling. Therefore, NPP is an indispensable data source when calculating GEP.

(3) Climate and Meteorological Data:

Relationship: Climate data includes factors such as temperature and precipitation, which have a significant impact on ecosystem productivity, water resource regulation, and more.

Importance: Climate data serves as a crucial basis for assessing ecosystem service values, directly related to various ecosystem services such as water resource supply, crop yields, and water resource regulation. Meteorological data can also be used to model the impact of climate change on ecosystem services.

(4) Statistical Yearbook Data:

Relationship: Statistical yearbooks contain various data related to the economy, society, and the environment, providing information about regional economic activities, GDP, employment, and more.

Importance: Statistical yearbook data offers critical information for evaluating the contribution of ecosystem services to the economy. By using statistical yearbook data, the economic value of ecosystem services can be linked to local economic activities, providing reference data for environmental policy development.

In summary, each data source plays a vital role in GEP calculations. They provide diverse information that helps us comprehensively understand the value of ecosystem services and connect it to economic activities, thus providing a scientific basis for sustainable development.  (See Page 3, Line124-131;)

In the sentence "Building upon previous research and on-site investigations," it would be helpful to provide a brief reference to or citation of the relevant previous research.

Response: Thanks for your proposal. It has been changed.  (See Page 5, Line 141;)

It might be beneficial to provide a brief rationale for why spatial autocorrelation analysis is relevant to your study. How does it contribute to understanding the spatial patterns of GEP or other aspects of the research?

Response: Spatial autocorrelation analysis helps to reveal the spatial distribution patterns of GEP, identify spatial dependencies, assess statistical significance, and provides policymakers with deeper insights into environmental and societal issues. Through this analysis, researchers can gain a better understanding of GEP and its relationship with geographic factors, enabling them to address relevant issues more effectively. (See Page 11-12 , Line 253-257 ;)

The analysis of temporal changes in GEP is valuable. However, some explanations of the findings would enhance the reader's understanding. For instance, you mention that "water conservation showed the highest rate of change, reaching 69.17%." Providing some context as to why this occurred or its implications for the environment would be beneficial.

Response: Thanks for your proposal. It has been added. (See Page 8, Line 209-212 ;)

Ensure consistency in units throughout the tables. In Table 4, the unit "Value(million)" is not a standard unit of measurement, and it should be clarified.

Response: Thanks for your proposal. Value (Hundred million yuan)" can be translated to "Value (in hundred million yuan)" or "Monetary value (in hundred million yuan)"

While the use of Moran's Index and Local Moran I is appropriate, it's important to explain their practical significance. How do the values of these indices inform your study, and what do they suggest about the spatial distribution of GEP?

Response: These indices help determine whether the spatial distribution of GEP exhibits significant spatial patterns and whether these patterns occur randomly. They provide guidance to researchers for gaining deeper insights into the geographical distribution trends of GEP and offer direction to policymakers for taking targeted actions. For example, if clusters of high values are identified in certain regions, the government may consider implementing environmental conservation measures or resource management policies in those areas. Similarly, negative Local Moran I values may indicate the need for special attention to address potential issues in specific areas.

In Figure 6, where you present the change in the global Moran's index, it would be beneficial to clarify the significance of the increasing trend. Is this trend expected, and what does it imply for the distribution of GEP in the region?

Response: Thanks for your proposal. It has been added. (See Page10-11 , Line253-257 ;)

Reviewer 2 Report

Dear Authors,

Please find my comments and suggestions in the attached file!

Kind regards!

Author Response

Why was this particular data selected?

Response: Thanks for your proposal. These data serve as the foundational sources for GEP calculations as they are closely tied to numerous ecosystem services. By integrating economic statistical data with ecosystem services, we can quantify the ecosystem's economic contributions and provide a scientific basis for environmental protection policies. Furthermore, this approach helps individuals gain a more comprehensive understanding of the economic value of ecosystems, thereby advancing the realization of sustainable development.

Why were these indicators calculated?

Response: The GEP (Gross Ecosystem Product) accounting framework is categorized into three types of services: provisioning, regulating, and cultural services. Under these three service categories, various specific functions are identified. Provisioning services encompass product supply and water resource supply. Regulating services include water resource regulation, flood control, carbon sequestration and oxygen production, soil retention, water purification, air purification, and windbreak and sand fixation. Cultural services involve aesthetic landscapes.

GEP measures the economic value of the various ecological services provided by ecosystems. Its purpose is to quantify the ecosystem's contribution to the economy, providing a scientific basis for environmental protection policies and decisions related to sustainable development. GEP accounting includes multiple indicators for provisioning, regulating, and cultural services, aiming to comprehensively reflect the ecosystem's economic contributions. This not only aids decision-makers in gaining a more comprehensive understanding of the economic value of ecosystems but also provides a scientific foundation for the formulation of environmental protection policies and strategies for sustainable development. Furthermore, it helps raise public awareness of ecological conservation and advances the achievement of sustainable development goals(See Page 15, Line 134-144;).

How are they important?

Response: The significance of these ecosystem service indicators lies in their crucial role in the sustainable development of human society and the economy. This importance is manifested in several key aspects:

(1) Economic Foundation: Provisioning services, such as product supply and water resource supply, have a direct impact on economic development and human livelihoods. Food, timber, and other resources sourced directly from ecosystems serve as the foundational materials for economic growth.

(2) Environmental Regulation: The various functions of regulating services are essential for maintaining environmental balance and stability. For instance, water resource regulation and flood control contribute to the sustainable use of water resources, while carbon sequestration and soil retention are critical for climate and soil regulation.

(3) Environmental Protection: Services like water purification and air purification play a positive role in cleansing and improving the environment, safeguarding the quality of human life.

(4) Health and Well-being: Provisioning services and aesthetic landscapes within cultural services have a direct impact on people's physical and mental health and sense of well-being. Beautiful natural landscapes can enhance people's mood and quality of life.

(5) Social Stability: The stable supply of ecosystem services plays a crucial role in societal stability and sustainable development. For example, stable water resources and effective flood control can reduce the impact of natural disasters on society.

(6) Sustainable Development: The rational utilization and protection of ecosystem services are prerequisites for achieving sustainable development goals. By accounting for these services, environmental protection policies and sustainable development strategies can be formulated more scientifically.

Therefore, the importance of these ecosystem service indicators lies in their direct connection to the survival, development, and well-being of human society, forming the foundation of the interdependence and mutual promotion between ecosystems and the socio-economic system. (See Page 6, Line 146-149;)

In what units are their components calculated?

Response: Statistical data is standardized to a common unit of measurement. For subsequent calculations, mass is measured in tons, and value is measured in thousands of yuan.  collect land use data, remote sensing NPP (Net Primary Productivity) data, meteorological data, and statistical yearbook data, including data on annual output of agriculture, forestry, animal husbandry, and fishery, grain production and planting area, various water usage, reservoir capacity, etc. Utilize methods such as the precipitation storage method, mass balance principle, soil erosion equation, modified wind erosion model, and more to calculate the physical quantity of products and services provided by various ecosystems.

Introduce pricing mechanisms and employ methods like market value, replacement cost, opportunity cost, and others to calculate the value of products and services provided by various ecosystems, which is then used to compute the Gross Ecosystem Product (GEP).

Part of these answers should be presented in the Introduction section, and another part in theMaterials and Methods section! In Materials and Methods section should be presented in detail: all data used, the basic steps in data processing (statistical, geospatial, etc.), the output data, obtained after the processing and their measuring units, software used for the data processing, how do the output data interact and how they are implemented in the methodology?

Response: Thanks for your proposal. To eliminate differences in units and spatial reference, a preprocessing step is carried out on all data before calculations. This involves transforming the coordinate system of raster data, such as land use data, precipitation data, and Net Primary Productivity (NPP) data, into the GCS_WGS_1984 geographic coordinate system. Additionally, the spatial resolution of raster data is resampled to 1 km × 1 km.Statistical data is standardized to a common unit of measurement. For subsequent calculations, mass is measured in tons, and value is measured in thousands of yuan.Raster data is processed using the raster calculator tool in ArcGIS software to obtain the unit area mass of variables like precipitation and NPP. The areas of various ecosystems are then determined. Using this information as a basis, the Gross Ecosystem Product (GEP) accounting framework is employed to calculate the mass and value of various ecosystem services. (See Page3 , Line125-130;)

Results The Results section should present only the results obtained without an explanation of how they were obtained, without the introduction of new terms (hot/cold spots)! All terms used should be introduced and explained in the text before the Results section!

Response: Thanks for your proposal. It has been changed.

Discussion The Discussion section includes general texts (e.g. 271-293), for the most part unrelated, even contradictory.

For example: If GEP is a comprehensive ecological value indicator and can be directly employed in assessments, why do you need to adapt diverse research methodologies in the future? (Lines 294-297)

Response: Although GEP serves as a significant ecological value indicator, future research will continue to necessitate the adoption of diverse research methodologies to continually enhance the quality and comprehensiveness of studies, thereby facilitating a better understanding and evaluation of ecosystem services. This does not imply a diminishment in the value of GEP; rather, it emphasizes that in varying research contexts and with different research objectives, different methods can complement each other, collectively advancing the field of ecology.   (See Page 15-16;)

You then list aspects for future research without making it clear why exactly those aspects were chosen!

Response: Thank you very much for the expert's valuable feedback. The suggestions regarding future research directions are intended to emphasize the rationale behind our chosen research methods and objectives, aiming to enhance the scientific value and practical applicability of the research. Here are some additional explanations regarding the selection of future research directions:

Firstly, we emphasize the diversity of data sources because the assessment of ecosystem services typically requires the inclusion of multiple data types and sources. In broad ecosystem assessments, we may need to use generalized data to gain a wide-ranging overview. However, for more detailed local assessments, finer and more precise data are required to better understand the characteristics of local ecosystems. Therefore, the diversity of data sources is aimed at ensuring that our research is both broadly applicable and rich in detailed information.

Secondly, multi-scale research is chosen because the characteristics of ecosystem services and ecosystems themselves exhibit significant variations across different spatial and temporal scales. We opt for a multi-level analysis, spanning from micro to macro perspectives, to comprehensively understand the ecological evolution and developmental trends within the Three Northern Protection Forest Project Area. Specifically, we focus on how the evolution of societal human factors has led to an increase in land use intensity, resulting in the gradual transformation of natural ecosystems into semi-natural, semi-artificial, and even fully artificial ecosystems. This transformation affects the flow of matter and energy, consequently influencing the capacity of ecosystems to provide services. Therefore, multi-scale research is instrumental in gaining a better understanding of these complex interactions.

Lastly, emphasizing the diversity of ecosystem services is because different types of ecosystems provide different types of services, and their contributions vary. We highlight this point to better consider the diverse contributions of different ecosystems to economic and social development. This aids in formulating more comprehensive and targeted regional development and ecological conservation strategies to better meet the needs of different regions and communities.

we have chosen these future research directions to better address the complexity of our study and the challenges that lie ahead, in order to provide in-depth research and practical guidance for the development and ecological conservation of the Three Northern Protection Forest Project Area. The selection of these directions is grounded in the aim of enhancing the scientific quality and practical applicability of our research. (See Page 15-16;)

Line 20 -The Three-North Shelterbelt Project is the largest ecological engineering initiative to date -

Where? In China or in the World?

Response: Thanks for your proposal. It has been changed. (See Page 1,Line 20-21;)

Line 23 -the Three Northern Protection Forest Project Area -Is this area part of the Three-North Shelterbelt Project? It is not clear!

Response: It has been changed. (See Page 3, Line 103-120;)

Line 50 -most challenging natural conditions -Most challenging for what?

Response: It has been added. (See Page 2, Line 50-52;)

Lines 86-91 -Reference missing!

Response: It has been added. (See Page 2, Line 86-91;)

Line 124 -Figure 1 -This figure is of low quality. Please, redraw it, so all text and numbers are readable!

Response: It has been changed. (See Page 3, Line 121;)

Line 128 -Table 1 -Geographic vector -What data does the geographic vector include?

Statistical yearbook -What data does the statistical yearbook include?

Response: Geospatial data includes administrative boundary data sourced from the Chinese Academy of Sciences Resource, Environment and Data Center. agriculture, forestry, animal husbandry, fisheries, grain production, planting area, water usage, and reservoir capacity. This data primarily originates from publications such as the "China Statistical Yearbook" and the "China Water Resources Statistical Yearbook." Additionally, specific regional data is obtained from "Water Resource Bulletins" from various areas within the research zone, with statistics collected from sources including the National Bureau of Statistics, local water resources departments, local forestry and grassland bureaus, local cultural and tourism bureaus, and other relevant entities.

Line 133-Figure 2-What kind of data includes the socio-economic data?

In addition, you have to include an explanation of why you chose exactly these indicators to calculate the GEP! This should be placed in the Introduction section!

Response: Socio-economic data comprises statistics related to agriculture, forestry, animal husbandry, fisheries, grain production, planting area, water usage, and reservoir capacity. This data primarily originates from publications such as the "China Statistical Yearbook" and the "China Water Resources Statistical Yearbook." Additionally, specific regional data is obtained from "Water Resource Bulletins" from various areas within the research zone, with statistics collected from sources including the National Bureau of Statistics, local water resources departments, local forestry and grassland bureaus, local cultural and tourism bureaus, and other relevant entities.

Socio-economic data includes statistical information related to agriculture, forestry, animal husbandry, fisheries, grain production, planting area, water usage, reservoir capacity, and more. Using socio-economic data for GEP calculations can provide governments and research institutions with a comprehensive and objective assessment tool. It assists them in better balancing the relationship between economic development and environmental conservation, ultimately helping to achieve sustainable development goals.

Line 147-2.3. Spatial autocorrelation of the Three-North Shelterbelt Project Area-What information did you gain after this analysis? Why is this calculation important?

Response: Conducting spatial autocorrelation analysis in the Three-North Protection Forest Project Area provides insights into the spatial distribution patterns of ecosystem service values among different locations within the region. The significance of this analysis lies in its ability to help us understand the spatial trends in ecosystem services across the geographic landscape. This understanding is crucial for more effective conservation and management of natural resources to meet both societal and economic needs.

Lines 166-181-e.g. 525.653 billion yuan -It is not necessary to mention all these values as they are presented in Table 3! Please, retain only the percentages!

Response: It has been changed. (See Page 6, Line 177-185;)

Lines 230-231-The green line represents the east-west trend, while the blue line represents the north-

south trend -Please, check it again!

Response: ok

Lines 247 -249 -This is for the Methodology section!

Response: It has been changed.

Line 266-Please, explain what do hot and cold spots mean and how the confidence values were calculated in the text! Such explanations should be placed in the Methodology section!

Response: Spatial analysis involves identifying significant hotspots and coldspots, which are phenomena or events in geographic space that exhibit significantly higher or lower values than the average. This analysis is commonly used to identify concentrated or dispersed patterns within a specific geographical area. To calculate confidence values for hotspots and coldspots, spatial statistical methods are typically employed. One commonly used method is the Getis-Ord Gi* statistic, also known as hotspot analysis or G-statistics. Statistical significance is determined using a p-value (significance level), with conclusions drawn if the p-value is less than the chosen significance level, typi-cally 0.05.This study employed the Anselin Local Moran I tool in ArcGIS 10.7 and overlaid land use change maps. (See Page 6, Line 165-174;)

Reviewer 3 Report

Overall, I think the research is important and scientifically sound. The authors did a fantastic job. However, I expect the authors to address my comments line-by-line. For any rebuttals, I need a detailed or comprehensive explanation.

Minor editing required with the punctuations, etc. (see my comments).

Author Response

TITLE: Can you rewrite the title as: "Conservation Effectiveness Assessment of the Three Northern Protection Forest Project Area."

Response: Thanks for your proposal. It has been changed. (See Page , Line 2-3 ;)

ABSTRACT: The abstract is well-written, but I recommend the following changes: Line 32: put a semi-colon after the word "function"and write as: "..... environmental purification functions; (ii) The per-unit area value of different.....

Response: Thanks for your proposal. It has been changed. (See Page 1, Line 20-34 ;)

Line 34: put a semi-colon after "49.84%"and write: "by shrubland ecosystems at 49.84%; (iii) The spatial distribution of ecosystem.

Response: Thanks for your proposal. It has been changed. (See Page 1, Line 32-34 ;)

  1. INTRODUCTION: Well written and cited section, but I suggest the following changes: Line 60-61: Please cite this sentence, may be use citations 2 and 3, if applicable: "It has enhanced the ecological environment, effectively addressing wind erosion and sand fixation, soil conservation, and water retention [2,3]."

Response: Thanks for your proposal. It has been changed.  (See Page 2, Line 50-52;)

Line 105: put a semi-colon after the word "construction"and write as: "..... construction; (ii) Evaluate and analyze the evolving characteristics and trends of the Gross Ecosystem Product (GEP) calculation results within the Three Northern Protection Forest Project Area."

Response: Thanks for your proposal. It has been changed. (See Page3, Line104-105;)

  1. MATERIALS AND METHODS: Looks good, but I recommend the following changes: Line 128 : In Table 1, authors mention "National Bureau of Statistics"do you have a link/website you can share like you did for the other classifications (for example, Land use data: https://www. resdc. cn).

Response: Thanks for your proposal. It has been changed.  (See Page 3-4, Line 124-131;)

Line 133: In Figure 2, can you make some of the variables more visible for the reader. For example,

"Dynamic Changes."

Response: Thanks for your proposal. It has been changed.  (See Page 4, Line 131-132;)

  1. RESULTS: Looks good, but I recommend the following changes: Line 171-173: rewrite sentence like this: "In terms of ecosystem types, grassland ecosystems possessed the highest value at 1,378.266 billion yuan, accounting for 68.81%, followed by farmland ecosystems at 338.983 billion yuan, making up 16.92%."

Response: Thanks for your proposal. It has been changed. (See Page 6 , Line177-180;)

Line 191: For Table 3, I recommend putting the Total Values (1507.88; 766.25; 13782.66; 3389.83;

583.43; 20030.05) in bold.

Response: Thanks for your proposal. It has been changed.  (See Page 7, Table 3;)

Line 224 & 225/226: Again, I recommend authors to put all the total values in bold.

Response: Thanks for your proposal. It has been changed. (See Page 9, Line 233-236;)

Line 229: Can the authors tell me the difference between the Figures (a-e)?

Response: Thanks for your proposal. The right side of the green line is ascending, and the Gross Ecosystem Product on the east side is increasing year by year.

Round 2

Reviewer 1 Report

Presented manuscript has significantly improved from the last revision and in my opinion can be published in present form

Author Response

Thank you 

Reviewer 2 Report

Dear Authors,

Thank you for your responses! They are comprehensive and detailed.

I am aware of all of that that I have asked in my first review report! I have asked all these questions so you can find out what is missing as information in your MS. Not every reader is familiar with the paradigm of ecosystem services, their importance, types, how they are valued, etc.! For that reason, it is important all responses to be addressed not to me but to the potential readers! Please include all these explanations with the appropriate references in the text where they fit best (Introduction or  Methodology section)! I have guided you about this in my first report.

The Discussion section does not consider results achieved by other authors in the field, which confirm or not the results obtained in the present study! Authors should discuss the results and how they can be interpreted in perspective of previous studies and of the working hypotheses. The findings and their implications should be discussed in the broadest context possible and limitations of the work highlighted.

The Conclusion section should be after the Discussion!

Kind regards!

Author Response

Response: Thank you for your detailed feedback and valuable insights. We have carefully reviewed your comments and made corresponding revisions to ensure that our research better meets the needs of potential readers.In response to your feedback, we have included more explanations about the paradigm of ecosystem services, their importance, types, and how valuation is conducted in the Introduction and Methodology sections. This is aimed at helping readers who may not be familiar with this field to better understand our study(See Page 4, Line 133-153;).

Regarding the Discussion section, we have considered previous research in the field and compared it with our study's findings, providing a comprehensive understanding of our research results and placing them within the context of prior research and hypotheses. Additionally, we have emphasized the limitations of the study to provide a more thorough discussion(See Page 14, Line 332-353;).

Finally, concerning the placement of the Conclusion section after the Discussion, we have made the necessary adjustments in line with your suggestion(See Page 15;).

We sincerely appreciate your review and guidance, as it will contribute to improving the quality and comprehensibility of our research. If you have any further suggestions for modifications or any other aspects, we are very eager to hear them.

Once again, thank you for your valuable feedback.
